# Demystifying excessively volatile human learning: A Bayesian persistent prior and a neural approximation

**Chaitanya K. Ryali**
Department of Computer Science and Engineering
University of California San Diego
9500 Gilman Drive La Jolla, CA 92093
rckrishn@eng.ucsd.edu

**Gautam Reddy**
Department of Physics
University of California San Diego
9500 Gilman Drive La Jolla, CA 92093
gnallama@physics.ucsd.edu

**Angela J. Yu**
Department of Cognitive Science
University of California San Diego
9500 Gilman Drive La Jolla, CA 92093
ajyu@ucsd.edu

## Abstract

Understanding how humans and animals learn about statistical regularities in stable and volatile environments, and utilize these regularities to make predictions and decisions, is an important problem in neuroscience and psychology. Using a Bayesian modeling framework, specifically the Dynamic Belief Model (DBM), it has previously been shown that humans tend to make the *default* assumption that environmental statistics undergo abrupt, unsignaled changes, even when environmental statistics are actually stable. Because exact Bayesian inference in this setting, an example of switching state space models, is computationally intensive, a number of approximately Bayesian and heuristic algorithms have been proposed to account for learning/prediction in the brain. Here, we examine a neurally plausible algorithm, a special case of leaky integration dynamics we denote as EXP (for exponential filtering), that is significantly simpler than all previously suggested algorithms except for the delta-learning rule, and which far outperforms the delta rule in approximating Bayesian prediction performance. We derive the theoretical relationship between DBM and EXP, and show that EXP gains computational efficiency by foregoing the representation of inferential uncertainty (as does the delta rule), but that it nevertheless achieves near-Bayesian performance due to its ability to incorporate a "persistent prior" influence unique to DBM and absent from the other algorithms. Furthermore, we show that EXP is comparable to DBM but better than all other models in reproducing human behavior in a visual search task, suggesting that human learning and prediction also incorporates an element of persistent prior. More broadly, our work demonstrates that when observations are information-poor, detecting changes or modulating the learning rate is both *difficult* and thus *unnecessary* for making Bayes-optimal predictions.

## Introduction

Understanding how humans and animals make future predictions based on changing environmental statistics is an important problem in both neuroscience and psychology [1, 2, 3, 4, 5, 6]. Intriguingly, even when environmental statistics are stable, Bayesian models of human learning and prediction suggest a default human tendency to assume that statistical contingencies undergo abrupt, unsignaled

changes, also known as "change points" [7, 8, 9, 10, 11, 12]. The behavioral consequence of this is that humans all too readily discard long-term knowledge in favor of recent, unexpected observations, leading to excessively volatile learning and prediction. It has been suggested that this default assumption of non-stationarity helps the brain to adapt when the environment is truly volatile [7]. Here, we propose another reason why a default assumption of volatility is difficult to overcome: one's ability to discern whether unexpected outcomes arise from change points or simply noise is fundamentally limited when the observations are very noisy (e.g. when it is binary as opposed to real-valued) [7]. We focus on categorical data (including binary) case, whose information-poor data (in a predictive information sense [13]) make the detection of change points and the estimation of hidden variables particularly difficult.

Previously, Bayesian and Bayes-inspired models of varying complexity have been suggested to capture human learning and prediction behavior while making implicit [3, 7] or explicit [1, 5] predictions among categorical choices. The most complex of these is exact Bayes [1, 3, 5, 7], such as the Dynamic Belief Model (DBM) [7], a hidden Markov model that assumes the observations to be drawn from a Bernoulli (if binary) [7] or categorical (if more than two outcomes) [11] distribution, whose parameters undergo abrupt, unsignaled changes from time to time. An alternative Bayesian model is the Fixed Belief Model (FBM) [7], which assumes environmental statistics to be *fixed* over time (no change points). It has been found that DBM captures human behavior better than FBM, even though the latter more veridically captures experimental design in a variety of tasks, e.g. 2-alternative forced choice [7], inhibitory control [9, 10], multi-armed bandit [12, 14], and visual search [11]. However, exact learning/prediction in DBM is computationally intensive, given that it is an example of switching state space models [15]. Consequently, several approximate and heuristic learning rules have also been proposed [1, 16, 17], all of which make some claim to neural plausibility and probabilistic interpretation. Separately, very simple, non-probabilistic forms of learning rules have also been used to model online learning in the brain. We explore two of them here: (1) a delta-learning rule [18, 19], also known as Q-learning or reinforcement learning (RL) in the neuroscience literature [5], (2) a variant of exponential filtering (EXP) [2, 7], equivalent to a particular form of leaky-integrating neuronal dynamics [7].

Although all of the algorithms described above have been used to model sequential learning and prediction in the brain, there has been little theoretical analysis of the statistical relationship among them, or a systematic validation by comparing them to the same set of behavioral data. In this work, we present just such a theoretical analysis and human data comparison [11].

The rest of the paper is organized as follows. In section 1, we will formally describe how the different algorithms learn online from binary data and make predictions about upcoming data. In section 2, we will present a theoretical analysis of the various algorithms and their relationships to each other. In section 2.5, we will extend the results to m-ary data. In section 3, we will compare model performance in terms of their ability to predict human behavior in a visual search task [11]. In section 4, we will discuss implications, links to related work, and future work.

# 1  Learning Models

In this section, we formally describe the learning models: the first two are principled Bayesian models, while the latter two are simple, mechanistic algorithms commonly used in neuroscience and psychology. Here, we assume the observations $x_t$ are binary. In a later section, we will show that our results easily generalize to the $m$-ary case.

## 1.1  Dynamic Belief Model (DBM)

The Dynamic Belief Model (DBM) is a hidden Markov model that assumes the observations are drawn from a Bernoulli distribution whose rate parameter undergoes unsignaled changes with probability $1 - \alpha$ at each time step.

**Generative Model.** The hidden variable $\gamma_t$ denotes probability of $x_t = 1$ and has a Markovian dependence on $\gamma_{t-1}$:

$$p(\gamma_t = \gamma | \gamma_{t-1}) = \alpha \delta(\gamma - \gamma_{t-1}) + (1 - \alpha)p_0(\gamma), \tag{1}$$

i.e., $\gamma_t$ remains the same ($\gamma_t = \gamma_{t-1}$) with a fixed probability $\alpha$, and redrawn from the prior $p_0(\gamma) = \text{Beta}(\gamma; a, b)$ with probability $1 - \alpha$.

**Recognition model.** The prior $p(\gamma_t|x_{1:t-1})$ and the posterior $p(\gamma_t|x_{1:t})$ are recursively computed:

$$p(\gamma_t = \gamma|x_{1:t-1}) = \alpha p(\gamma_{t-1} = \gamma|x_{1:t-1}) + (1-\alpha)p_0(\gamma_t = \gamma), \qquad (2)$$

$$p(\gamma_t|x_{1:t}) \propto p(x_t|\gamma_t)p(\gamma_t|x_{1:t-1}). \qquad (3)$$

**Prediction.** The predictive probability for trial $t+1$, given the past observations $x_{1:t}$ is computed as

$$P_{\text{DBM},t+1} \triangleq P(x_{t+1} = 1|x_{1:t}) = \int \gamma p(\gamma_{t+1} = \gamma|x_{1:t})d\gamma = E_{p(\gamma_{t+1}|x_{1:t})}[\gamma], \qquad (4)$$

and has an implicit marginalization over every possible timing of the most recent change point. In practice, one can either marginalize over the timing of the last change point, or discretize the belief state (posterior distribution over $\gamma_t$). Thus, the computation of the predictive probabilities is computationally and representationally expensive.

## 1.2 Fixed Belief Model (FBM)

FBM is a special case of the DBM with no change point, i.e $\alpha = 1$. It is simply a beta-Bernoulli process. The posterior and predictive probabilities are:

$$p(\gamma|x_{1:t}) \propto P(x_{1:t}|\gamma)p(\gamma) = \gamma^{\sum x_\tau + a - 1}(1-\gamma)^{\sum \bar{x}_\tau + b - 1}; \; P_{\text{FBM},t+1} \triangleq \frac{a + \sum x_\tau}{a + b + t}, \qquad (5)$$

where $\bar{x}_\tau \triangleq 1 - x_\tau$.

## 1.3 Exponential Filtering (EXP)

EXP is a simple algorithm that linearly sums past observations, while exponentially discounting into the past [2], to predict the probability of encountering different outcomes on the next trial [7]:

$$P_{\text{EXP},t+1} \triangleq P_{\text{EXP}}(x_{t+1} = 1|x_{1:t}) = C + \eta\beta \sum_{\tau=0}^{t-1} \beta^\tau x_{t-\tau} = C(1-\beta) + \eta\beta x_t + \beta P_{\text{EXP},t}, \qquad (6)$$

where the parameters $(C, \eta, \beta)$ are constrained as $0 \leq C, \eta \leq 1, 0 \leq \beta < 1, C + \frac{\eta\beta}{1-\beta} < 1$. This model was introduced in relation to DBM [7], inspired by related work showing that monkeys' choices when tracking reward biases that undergo change points are discounted in an approximately exponential fashion [2]. The last expression in Eq. 6 shows how it can be implemented by correctly tuned leaky integration dynamics (in a single neuron!) [7]: the first term is a constant bias, the second "feedforward" term depends on the current input ($\eta\beta$ specifies the weight on the input), and the third "recurrent" term depends on the previous state ($\beta$ specifies the weight of the recurrent term).

## 1.4 Delta-Learning Rule (RL)

The delta-learning rule, a form of simple Q-learning or reinforcement learning (RL) [19] is commonly used for online learning in both neuroscience [18, 5] and machine learning [19]. Here, we adapt it to estimate predictive probabilities:

$$P_{\text{RL},t+1} \triangleq \epsilon x_t + (1-\epsilon)P_{\text{RL},t}. \qquad (7)$$

Note that this version of RL is similar in form to EXP. It has a feedforward term and a recurrent term, one parameter to trade off between the two, and no bias term.

## 2 Relationship Among the Models

In this section, we analyze the relationship among the models. We will first show that while DBM online prediction can be viewed as a delta-like learning rule with an adaptive gain, and EXP, with a constant learning rate, can nevertheless approximate DBM well under certain conditions. We will also show when and why EXP outperforms RL, as well as how the parameters of EXP can be tuned online in a neurally plausible manner. Finally, we will analyze the parameter regime under which the DBM $\approx$ EXP approximation breaks down.

## 2.1 DBM Prediction as an Adaptive Delta Rule

The exact, nonlinear Bayesian update rule for the predictive probability $P_{\text{DBM},t+1}$, denoted as $P_{t+1}$ in this section to be concise, may also be written as:

$$P_{t+1} = (1-\alpha)\langle\gamma\rangle_{p_0(\gamma)} + \alpha x_t \frac{Q_t - P_t^2}{P_t(1-P_t)} + \alpha P_t \frac{P_t - Q_t}{P_t(1-P_t)} \tag{8}$$

$$= (1-\alpha)P_0 + \alpha x_t G_t + \alpha P_t(1-G_t) = (1-\alpha)P_0 + \alpha(P_t + G_t(x_t - P_t)), \tag{9}$$

where $Q_t \triangleq E_{p(\gamma_t|x_{1:t-1})}[\gamma^2]$, $P_0 \triangleq E_{p_0(\gamma)}[\gamma]$ and $G_t \triangleq \frac{Q_t - P_t^2}{P_t(1-P_t)} = \frac{\text{var}(\gamma_t|x_{1:t-1})}{\text{var}(\text{Bern}(P_t))}$. The form in Eq. (9) is reminiscent of the delta rule: $G_t$ ($0 \le G_t \le 1$, for any binary sequence $x_{1:t}$) acts like an adaptive learning rate, governing the trade-off between new data $x_t$ and the previous predictive mean, $P_t$; an additional parameter $\alpha$ governs the trade-off between this combined prediction and a constant bias $P_0$, which inserts *persistent prior influence* due to the recurring probability of $\gamma$ being re-sampled.

Intuitively, $G_t$ is modulated by how "surprising" recent observations are. Surprising recent observations, i.e those inducing large prediction error, could indicate a switch in environment statistics, prompting an increase in the learning rate. However, categorical data are information-poor, making prompt detection of a true change in the environment difficult. This suggests that the Bayesian update rule for predicting future outcomes can be simplified by approximating $G_t$ with an appropriate constant. These intuitions are formalized in the following theorem.

**Theorem 1.** *The adaptive learning rate $G_t$ has the following property,*

$$1 - G_t = (1 - G) + \alpha c_\alpha(-a\bar{x}_{t-1} + bx_{t-1}) + O(\alpha^2), \tag{10}$$

*where $G = \frac{1}{(a+b+1)}$ and $c_\alpha = \frac{(a^2-b^2)}{ab(a+b+1)^2(a+b+2)}$. Approximating $G_t$ by $G$ yields a linear update rule for the predictive probability $P_{t+1}$, correct to $O(\alpha^2)$,*

$$P_{t+1} = (1-\alpha)P_0 + \alpha(Gx_t + P_t(1-G)) + O(\alpha^2). \tag{11}$$

*Proof.* We rewrite the update rule (9) for the predictive probability $P_{t+1}$ as:

$$P_{t+1} = (1-\alpha)P_0 + \alpha(P_t + G_t(x_t - P_t)) = (1-\alpha)P_0 + \alpha L_t,$$

where $L_t \triangleq x_t G_t + P_t(1-G_t)$. Analogous to the update rule for $P_t$ (Eq. 8), $Q_t$ has the update rule:

$$Q_{t+1} = (1-\alpha)Q_0 + \alpha x_t \frac{R_t - Q_t P_t}{P_t(1-P_t)} + \alpha Q_t \frac{Q_t - R_t}{Q_t(1-P_t)}, \tag{12}$$

where $R_t \triangleq E_{p(\gamma_t|x_{1:t-1})}[\gamma^3]$. Next, we make $O(\alpha^2)$ approximations to the numerator $(P_t - Q_t)$ and the denominator $P_t(1-P_t)$ of $1 - G_t$ as

$$P_t(1-P_t) = P_0\bar{P}_0 + \alpha[\bar{P}_0 L_{t-1} + P_0\bar{L}_{t-1} - 2P_0\bar{P}_0] + \alpha^2(P_0 - L_{t-1})(\bar{P}_0 - \bar{L}_{t-1})$$

$$\overset{(*)}{=} P_0\bar{P}_0 + \alpha\frac{(a-b)}{(a+b)^2(a+b+1)}(-a\bar{x}_{t-1} + bx_{t-1}) + O(\alpha^2), \tag{13}$$

$$P_t - Q_t \overset{(*)}{=} (P_0 - Q_0) - \alpha\frac{(a-b)}{(a+b)(a+b+1)(a+b+2)}(-a\bar{x}_{t-1} + bx_{t-1}) + O(\alpha^2), \tag{14}$$

where $\bar{P}_0 = 1 - P_0$ and $\bar{L}_{t-1} = 1 - L_{t-1}$ and $(*)$ follows by setting $P_{t-1} = P_0 + O(\alpha)$, $Q_{t-1} = Q_0 + O(\alpha)$, $R_{t-1} = R_0 + O(\alpha)$. Upon substituting the approximations (13), (14) for $(P_t - Q_t)$ and $P_t(1-P_t)$ and using $(l_0 + \alpha l_1 + O(\alpha^2))^{-1} = l_0^{-1} - \alpha l_1 l_0^{-2} + O(\alpha^2)$, the $O(\alpha^2)$ approximation (10) for $G_t$ directly follows.

Setting $G_t = G + O(\alpha)$ in (9) gives (11), the linear update rule for the predictive probability $P_{t+1}$ correct to $O(\alpha^2)$. □

Based on the theorem, $G_t$ can be approximated as a constant with $O(\alpha)$ error, or as a linear function of the last observation with $O(\alpha^2)$ error; the corresponding linear update rule has either $O(\alpha^2)$ or $O(\alpha^3)$ error, respectively. Furthermore, $|c_\alpha(-a\bar{x}_{t-1} + bx_{t-1})|$ can be shown to be upper bounded

by a small number of 0.062 (proof omitted) for $a, b \geq 1$, so replacing $G_t$ by $G$ should work well in practice.

As a corollary to the theorem, for a uniform prior $a = b = 1$, the $O(\alpha)$ term in $1 - G_t$ is exactly zero, so that replacing $G_t$ by $G$ incurs only $O(\alpha^3)$ error. In many behavioral tasks (e.g. 2-alternative forced choice), a uniform prior is a reasonable choice; we employ a uniform prior for all simulations in the paper. All these results imply that the approximations are particularly accurate when $\alpha$ is relatively small.

On a separate note, it is important to note that the proof, and thus the approximation, does not make any specific generative assumptions about the sequence $x_{1:t}$, and is therefore valid for *arbitrary* binary sequences. In other words, the constant $G_t$ approximation is valid for arbitrary environments and does not depend on whether humans truly have mis-specified generative assumptions, as having been previously suggested [7, 11, 12].

## 2.2   Relationship of EXP to DBM and RL

We define EXP using Eq. 11. We will show that while two critical features of DBM – exponential discounting of past observations and "persistent" influence of the prior – are captured by EXP, only the former is captured by RL. Moreover, in volatile environments (relatively small $\alpha$, which appears to be the default assumption for humans, see sec. 4), EXP will be shown to be especially effective at approximating DBM, while also enjoying a particular advantage over RL.

Eq. 11 shows how the parameters of EXP are related to those of DBM: $\beta = \alpha\frac{a+b}{a+b+1}$, $\eta = \frac{1}{a+b}$, $C = \frac{(1-\alpha)P_0}{1-\beta}$. In other words, the exponential discount parameter of EXP, $\beta$, is proportional to the volatility parameter $\alpha$ in DBM (for uniform prior, $\beta \approx \frac{2}{3}\alpha$, matching a previous conjecture [7]), and the constant bias, $(1-\alpha)P_0$, is proportional to the prior mean $P_0$ and thus injects a persistent additive influence of the prior. In a set of simulations with $\alpha = .7$ (similar to those found in humans, see sec. 4), we regress $P_{\text{DBM},t+1}$ against past observations $x_t, x_{t-1}, \ldots$, and find that our analytical EXP approximation closely matches both DBM and the best freely fitted EXP (best linear estimator) (Fig. 1a). We also see that this excellent performance is underpinned by an approximately constant learning rate $G_t$ that is quite insensitive to the timing of true change points (Fig. 1b). Indeed, EXP approximates DBM equally well whether there is a switch on the last time step or not (Fig. 1c).

We can gain additional intuition about DBM (and EXP) by noticing that the parameter $C$ is the lower bound on $P_t$, determined by the stability of the environment $\alpha$ and the prior $p_0(\gamma)$. This lower bound is attained asymptotically in the limit of observing an infinite sequence of 0's. Similarly, the upper bound in the limit of observing infinite 1's is $C + \frac{\eta\beta}{1-\beta}$ (see the last ten trials in Fig. 1a). This bounded behavior is characteristic of DBM, and well captured by appropriately parameterized EXP.

Like EXP, RL has an element of exponential discounting if we write $P_{\text{RL},t+1} = \sum_{\tau=0}^{t-1} \epsilon(1-\epsilon)^\tau x_{t-\tau}$. However, RL has no analytic setting for its free parameter $\epsilon$, and parameter fitting yields a discount behavior different from DBM and EXP (Fig. 1d). An even bigger problem is that RL cannot capture a persistent prior influence, due to the lack of a bias term. In a small-$\alpha$ environment, the persistent prior influence is especially critical (Eq. 9), and EXP enjoys a particular advantage over RL (Fig. 1a). This pattern also translates to the behaviorally more relevant measure of predictive accuracy (Fig. 1e), which assumes the observer to make a binary outcome prediction choice (by taking the max) based on predictive probabilities.

It is worth noting that even in this regime of relatively frequent switches, prediction is not trivial, in that it depends sensitively on the data and not only the prior (Fig. 1a;d); a prediction algorithm that only relies on the prior performs poorly (Fig. 1e). Indeed, smaller $\alpha$ makes DBM and EXP especially sensitive to local statistics (recent data), since they are more willing to discard long-term statistics due to the stronger belief that environmental statistics can drastically change any time.

## 2.3   Adapting to Volatility

Humans appear to be able to adapt their choice behavior according to changing volatility $(1 - \alpha)$ of the environment [5]. Exact Bayesian is computationally intensive. However, the EXP approximation, denoted $\hat{P}_t$, to the DBM permits a simple, principled update rule for $\alpha$ via stochastic gradient descent

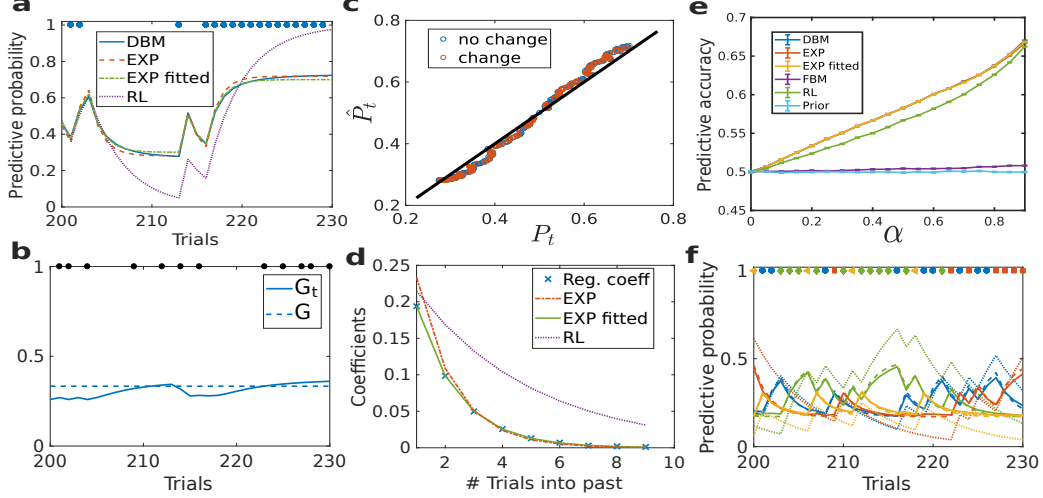

Figure 1: Simulation results: validity of EXP approximation. Data generated from DBM ($\alpha = 0.7$) (a-d): $m = 2$ (binary data), $p_0(\gamma) = \text{Beta}(1,1)$. (a) Exact and approximate predictive probabilities (of observing 1) for an example sequence of synthetic data (1's depicted by blue dots, 0's not shown). (b) Exact $G_t$ and approximate $G$ learning rates of an example sequence; black dots denote true change points. (c) Approximate predictive probability $\hat{P}_t$ (EXP) versus exact $P_t$ (DBM), following no change point (blue) or a change point (red). (d) DBM dependence on previous observations (blue: linear regression coefficients) is approximately exponential (green), and well-approximated by EXP (red). Fitted RL yields a very different exponential curve (purple). **e** Predictive accuracy (fraction of correct predictions): DBM≈EXP≈EXP fitted > RL > FBM. (f) Analogous to (a) but for $m = 4$ and $p_0(\gamma) = \text{Dir}(1,1,1,1)$. Different colors represent the four outcomes.

[7]:

$$\hat{\alpha} \leftarrow \hat{\alpha} + \epsilon(x_t - \hat{P}_t)\frac{d\hat{P}_t}{d\hat{\alpha}}; \ \frac{d\hat{P}_t}{d\hat{\alpha}} = \hat{P}_{t-1} + G(x_t - \hat{P}_{t-1}) - P_0. \tag{15}$$

### 2.4 Breakdown of DBM ≈ EXP

For $\alpha \approx 1$, EXP is not a good approximation to exact-Bayes predictive probabilities (Fig. 2a). Indeed, for a stable FBM environment ($\alpha = 1$), $G_t = \frac{1}{a+b+t}$, which is clearly not constant. However, fitting EXP's parameters freely still performs close to DBM (Fig. 2a); while the deviation between our analytical approximation of the discount parameter $\beta$ and the best fitting $\beta$ grows as a function of $\alpha$ (Fig. 2c). For larger $\alpha$, even though $G_t$ increases more after a true change point, because real change points are *rare*, their influence is minor relative to the stable value of $G_t$ in between change points. In any case, in terms of the behaviorally more relevant *predictive accuracy* measure (analogous to Fig. 1e), EXP still approximates DBM well (Fig. 2b). Interestingly, fitted RL also approximates DBM well (Fig. 2a;b), since the persistent prior influence in Eq. 11 is more negligible. This makes a broader point about prediction in stable but noisy environments: simple, cheap prediction algorithms can perform well relative to complex models, since each data point contains little information and there are many consecutive opportunities to learn from the data.

### 2.5 Generalization to $m$-ary Data

DBM and EXP easily extend to $m$-ary data. We assume a Dirichlet prior $p_0(\gamma) = \text{Dir}(\mathbf{a})$, where $\mathbf{a} = (a_1, \ldots, a_m)$, $a_k \geq 1$. We say $x_t^{(i)} = 1$ if the observation on trial $t$ is category $i$. Denoting $P_{t+1}^{(i)} \triangleq P(x_{t+1}^{(i)} = 1|x_{1:t}^{(i)})$ and $a_{-i} \triangleq \sum_{k \neq i} a_k$, the following corollary is easy to show (proof omitted):

**Corollary 1.** *The adaptive learning rate $G_t$ has the following property,*

$$1 - G_t = (1 - G) + \alpha c_\alpha^{(i)}(-a_i\bar{x}_{t-1}^{(i)} + a_{-i}x_{t-1}^{(i)}) + O(\alpha^2), \tag{16}$$

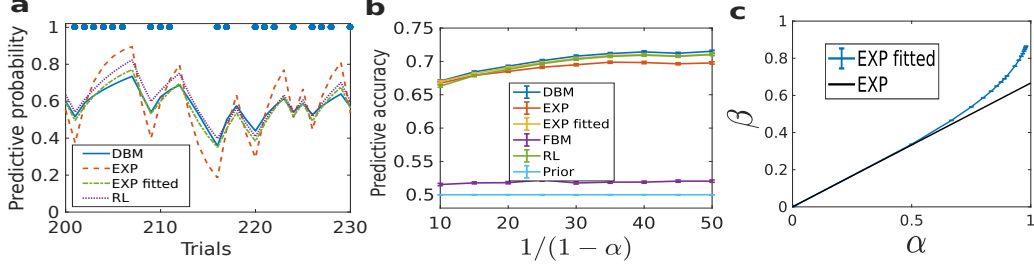

Figure 2: Simulation results: large $\alpha$. DBM parameters: $m = 2, p_0(\gamma) = \text{Beta}(1, 1)$. (a) Exact and approximate predictive probabilities, analogous to Fig. 1a. $\alpha = 0.95$. (b) Predictive accuracy, analogous to Fig. 1e, on the scale of $1/(1-\alpha)$ instead of $\alpha$ to better visualize performance for large $\alpha$. (c) EXP fitted parameters $\beta$ deviates from our approximation for larger $\alpha$. Comparison of analytical approximation of $\beta$ versus best fitted $\beta$, as a function of $\alpha$.

*where* $G = \frac{1}{(\sum_k a_k + 1)}$ *and* $c_\alpha = \frac{(a_i^2 - a_{-i}^2)}{a_i a_{-i} (\sum_k a_k + 1)^2 (\sum_k a_k + 2)}$. *Approximating* $G_t$ *by* $G$ *yields a linear update rule for the predictive probability* $P_{t+1}^{(i)}$, *correct to* $O(\alpha^2)$,

$$P_{t+1}^{(i)} = (1-\alpha)P_0^{(i)} + \alpha(Gx_t^{(i)} + P_t^{(i)}(1-G)) + O(\alpha^2). \tag{17}$$

Though we maintain approximate updates for each $P_t^{(i)}$ separately, it is easily shown by induction that normalization is preserved, $\sum_{i=1}^m P_t^{(i)} = 1$. Since identical bounds on the coefficient of $O(\alpha)$ term in $G_t$ hold for $m > 2$, the quality of the approximation will be the same as the binary case, so that in volatile environments, near-Bayesian prediction can be achieved simply by using $m - 1$ separate linear-exponential filters with no recurrent or complex interactions among the alternatives. We will use this novel $m$-EXP model in section 3 to model human data.

## 3 Case Study: Visual Search Task

We will evaluate the models by comparing the models to human behavior in a visual search task [11]. The objective of the task is to find the target among three stimuli (target is a random-dot patch moving in the direction opposite to the two distractor patches, see Fig. 3a). The location of the target on each trial is drawn independently from a fixed distribution (1/13, 3/13, 9/13). We collapse the spatial configuration and refer to the patches corresponding to prior probabilities 1/13, 3/13, 9/13 as patches 1, 3 and 9, respectively. The spatial configuration is fixed in a block (90 trials per block), and counter-balanced across blocks for each subject. Eye-movements are tracked; we only analyze first-fixation location here, as an indication of where a subject perceives as the currently most probable target location. Subsequent fixations are much more complex complex, being "contaminated" by sensory and motor processes [20]. Subjects are given feedback of true target location on each trial. The data are from 11 subjects and are from [11].

### 3.1 Model Fitting

Learning of environmental statistics by the participants is modeled using each of DBM, EXP, RL and FBM. DBM and FBM both assume an uninformative prior $p_0(\gamma) = \text{Dir}(\gamma; 1, 1, 1)$. Since the actual spatial configuration is fixed over a block, FBM is the correct generative model. The probability of the first fixation choice (*choice fraction*) $q_{t,i}$ at time $t$ is modeled by polynomial softmax [4, 11] as

$$q_{t,i} = \frac{(P_{t+1}^{(i)})^\beta}{\sum_i (P_{t+1}^{(i)})^\beta}.$$

We fit the learning and decision making models at an individual level by maximizing the likelihood of first fixation choices (averaged over trials). Each of DBM, EXP and RL have one free parameter ($\alpha$ for DBM/Exp, $\epsilon$ for RL), while FBM has none. The learning rate $G$ in EXP is set to $\frac{1}{(\sum a_k + 1)} = \frac{1}{4}$ according to the main theorem.

## 3.2 Results

As shown in [11], the aggregate choice statistics appear to correspond to matching [21] but belie the more complex temporal patterns in choice behaviour. In Fig. 3b, note that when the previous target was 1 or 3, the first fixation choice fractions on the next trial show a much higher choice fraction of 1 or 3, respectively. This bar graph is re-plotted in a different representation in Fig. 3c, where each choice distribution is represented by a point in this 2D probability simplex (2D because the three probabilities add up to 1), affine-transformed to achieve symmetry across the three choices. We see that, in comparison to the case when last target was location 9, human choice fractions on the current trial are pulled toward 1 or 3, when the last target was 1 or 3, respectively. DBM and EXP are biased to a similar extent. However, FBM, which asymptotically ignores the last data point, shows very little variation in average choice distribution as a function of last trial location. RL, which shares the exponential discounting element of DBM and EXP but not the persistent prior component, exhibits some influence of the last target location, but not as much as humans/DBM/EXP. Note that all model results are on held out data, and therefore independent of model complexity.

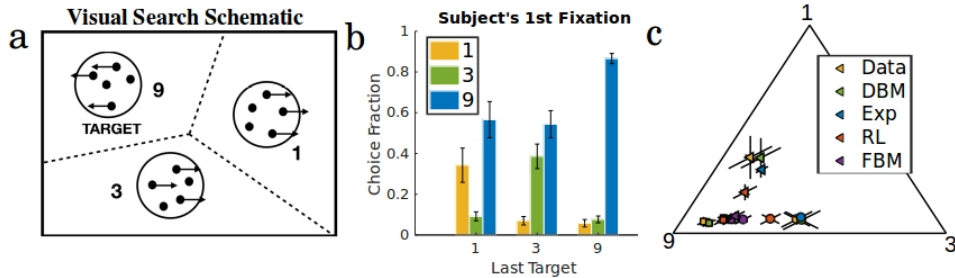

Figure 3: Model comparison to human data in visual search task. (a) Schematic of the task. (b) Human choice fractions conditioned on the last target location. (c) Model-predicted choice fractions and human choice fractions, on an affine transformation of the probability simplex. Last target location: 1 - △, 3 - ○, 9 - □. Model predictions are based on actual sequences of held out stimuli subjects experienced in 6-fold maximum likelihood cross validation. Error bars = *SEM* over subjects.

## 4 Discussion

We have shown that the DBM-like human learning/prediction found in previous studies can be implemented by appropriately tuned leaky integrating neuronal dynamics (EXP). While we derived an analytical form for the appropriate EXP parameters for volatile environments, we have also shown that even for less volatile environments, where our analytical approximation does not hold, the empirically fitted EXP still achieves near-Bayes performance. This leaves open the possibility that the brain may utilize EXP-like learning for quite a large range of possible volatility, via feedback-driven incremental tuning. In any case, in previous tasks where human behavior has been shown to be fitted well by DBM, fitted $\alpha$ ranges between 0.7 and 0.8 [7, 11, 12, 9, 10]) – in the range where our analytically derived EXP would perform very close to Bayes-optimal.

Our work demystifies human learning [7, 14, 11, 12, 9, 10] by decomposing DBM into two simple mechanistic components. We showed that EXP approximates DBM well in all but extremely stable environments, and does so via both exponential discounting of past observations, and a persistent influence of a prior bias that is injected on every trial. Our work shows that when observations are information-poor, detecting changes or modulating the learning rate explicitly or implicitly (e.g. by discretizing belief state space [7] or averaging over possible change point times [22, 17]) is both *difficult* and (thus) *unnecessary* for making Bayes-optimal predictions. In practice, $m$-ary DBM is typically implemented via discretization of the belief state space [7, 11, 9, 10], which has a computational and representational complexity of $O(e^{km})$ per observation, where $k$ depends on fineness of the discretization, while the near exact-Bayes approximation EXP is only $O(m)$.

We found that DBM & EXP both explain human choice behavior in a visual search task [11] better than RL, which has exponential discounting but no persistent prior influence, and FBM, which has neither. This is broadly consonant with our related finding that DBM not only provides a better trial-by-trial account of human-choices in a multi-arm bandit task, but is able to recover a systematic

underestimation in human prior reward rate expectation – this "pessimism bias" is incompletely captured by RL and FBM [14]. Together, these findings suggest that a more comprehensive comparison of these models in their ability to capture diverse behavioral patterns is needed in the future.

Note that we are not suggesting that humans can *only* do prediction with a constant learning rate. In "information-abundant" settings, change-points are relatively easy to detect, and their detection is critical for Bayes-optimal learning and prediction. We have done separate simulations (data not shown) to show that, in comparison to binary or categorical data, when mutual information between hidden state and observations is high, Bayesian detection of change points can be highly accurate, and the corresponding "learning rate" of its equivalent leaky-integrating update equation significantly increases after detecting such a change. In these scenarios, the EXP approximation with a constant learning rate would clearly do a poor job. Indeed, there is evidence that in information-abundant settings, human learning rate may be modulated by uncertainty [16], and subjects are able to detect change points and report uncertainty [23]. It is quite possible that different parts of the brain may implement different kinds of learning/prediction algorithms. Different approximations may come into play depending on information-abundance or whether the task explicitly necessitates the representation of uncertainty (e.g. in [23]).

Given how well EXP does as an approximate recognition model for DBM, and EXP does not bother to detect change points or modify its learning rate in response to detected change points, there might exist a generative model for which EXP would be an exact Bayesian recognition model. In particular, one might consider a model that assumes the underlying real-valued hidden variable to undergo persistent stochastic changes with constant noisy characteristics, such as a Gaussian process, which then gives rise to noisy binary or categorical observations. Finally, we note that our approximation technique does not preclude an approximation in which the learning rate is modulated from trial to trial. In fact, a proof technique similar to the one used here may be utilized to determine an approximate update rule for higher order moments of $p(\gamma_{t+1}|x_{1:t})$ (proof not shown), which could be used as a neurally plausible approximation to confidence in information-abundant settings. Whether such an approximation can account for human reported confidence as in [23] is a worthy line of inquiry for future work.

### Acknowledgments

We thank He Huang for assistance with data collection, Samer Sabri for helpful input with the writing, and the anonymous reviewers for helpful comments. This work was in part funded by an NSF CRCNS grant (BCS-1309346) to AJY.

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
