[Reviews · NeurIPS 2018]

Reviewer 1



This paper builds on a successful line of research from Yu and colleagues on change point detection. The paper presents some interesting theoretical results linking the Bayes-optimal solution to computationally efficient and neurally plausible approximations. The paper also presents a cursory analysis of empirical data using the approximations. The paper is well-written and technically rigorous. There were a number of important useful insights from the theoretical analysis. I was somewhat disappointed by the empirical analysis, as discussed below. Specific comments: The simple delta rule model with fixed learning rate is a bit of a straw man. More relevant are variants of the delta rule that use an adaptive learning rate (see Sutton, 1992). I'm not sure what we learn from the observation that the best fitting parameters allows the EXP approximation to still approximate DBM well when alpha is close to 1. This would be useful if the agent could estimate those parameters from observations. Do the analytical results suggest a context-dependent division of labor between parallel learning systems, which could be tested experimentally? I'm thinking specifically of the statement at the end of section 2.4: "simple, cheap prediction algorithms can perform well relative to complex models, since each data point contains little information and there are many consecutive opportunities to learn from the data." For example, it would be interesting to parametrically vary alpha in an experiment and test whether different approximations came into play depending on the alpha value. I was hoping for more from the empirical analysis. Is there any positive evidence for the EXP approximation over DBM? For example, a formal model comparison result, or some qualitative pattern in the data that would indicate people are using EXP? As it currently stands, there isn't any clear evidence distinguishing EXP and DBM. Although the paper cites the work by Wilson, Nassar and Gold, there isn't any substantive discussion of their work, which is rather similar: they use a mixture of delta-rules to approximate the Bayes-optimal solution. Minor comments: p. 5: "Exact Bayesian" -> "Exact Bayesian inference" ------------------- I think the authors gave fairly reasonable responses to my comments. I already gave the paper a fairly high score, and I'm not inclined to raise it because the main issue I raised was the non-identifiability of the models based on the behavioral data, which the authors concede is true. Apart from that issue (which is a pretty big one) I think the paper may be good enough for NIPS.

Reviewer 2



The authors introduce an approximate model of the Bayesian dynamic belief model (DBM) that is based on exponential filtering and can produce similar results as the DBM but at lower computational costs and increased neural plausibility. I like the idea of defining more plausible solutions to the puzzle of volatililty in human behavior. Yet this paper somehow left me wanting more. Here are some suggestions on how to improve it: - as an author I always dislike reviews that say the present work isn't novel enough myself, but this paper really felt a little like that; I don't think that this is necessarily the authors fault per se and definitely doesn't justify rejecting the paper. However, I think the intro could motivate the models a little more. I think it would make sense to mention some cases where humans are volatile, how DBM captures this behavior and why more plausible implementations based on approximations are useful (for example, mentioning computational and neural constraints). - some minor comments: sometimes the quotation marks are inverted, sometimes not. some of the equations aren't numbered. - The experiment is a little opaque to me. I don't really understand what was done there. Is this the data from the Yu & Huang paper? How many participants were there? What was their task and how did they get rewarded? I'm not fully sure what Figure 1c means. I guess it shows that the predictions of DBM and Exp align with the data in terms of which patch is likely on average. Is this the only thing the model can capture here? Are there any inter-individual differences that the model can pick up on? Why is the prediction only about the first fixation? Couldn't this be captured already by accounting for some level of stickiness based on what was previously right? How does this relate to volatility of participants' belief if it will be the same option again? I think this part needs more explanation - In general, I was really excited about the paper when I saw the title, but then it is never really explained how this actually demystifies volatile learning. Doesn't the DBM model already demystify volatility and the current model is a more neurally plausible approximation to that model? My concern about this paper is that it might be a little too niche for the general NIPS community as it is about a more plausible approximation to a model that has been proposed before, so people would essentially have to know the prior model, too. I don't think that's a bad idea and incremental work should be encouraged of course, but then I would really try and emphasize how this goes beyond DBM, perhaps even thinking about a novel prediction it could generate. - equivalent to a particular form of leaky-integrating neuronal dynamics [19]. -> I think the general NIPS audience would need another sentence here to explain why this is important. - I wonder how exponential filtering relates to other filtering approaches such as Kalman filtering or Harrison Ritz' control theoretic model of learning in changing envrionments.

Reviewer 3



The submission could have been written with more clarity despite an interesting modeling framework. At times, the language and phrasing comes across more convoluted than how it could have been expressed. However, the proposed method is creative in its approach and could be a valuable contribution. Addressing constraints and concerns with the method proposed would have been welcome, along with suggestions for future research.